# Surveillance of Severe Acute Respiratory Infection and Influenza Vaccine Effectiveness among Hospitalized Italian Adults, 2021/22 Season

**DOI:** 10.3390/vaccines11010083

**Published:** 2022-12-30

**Authors:** Donatella Panatto, Alexander Domnich, Maria Chironna, Daniela Loconsole, Christian Napoli, Alessandra Torsello, Ilaria Manini, Emanuele Montomoli, Elena Pariani, Silvana Castaldi, Andrea Orsi, Giancarlo Icardi

**Affiliations:** 1Department of Health Sciences (DISSAL), University of Genoa, 16132 Genoa, Italy; 2Interuniversity Research Center on Influenza and Other Transmissible Infections (CIRI-IT), 16132 Genoa, Italy; 3Hygiene Unit, San Martino Policlinico Hospital-IRCCS for Oncology and Neurosciences, 16132 Genoa, Italy; 4Interdisciplinary Department of Medicine, University of Bari, 70124 Bari, Italy; 5Sant’Andrea Hospital, University La Sapienza, 00189 Rome, Italy; 6Department of Medical Surgical Sciences and Translational Medicine, “Sapienza” University of Rome, 00189 Rome, Italy; 7Department of Molecular and Developmental Medicine, University of Siena, 53100 Siena, Italy; 8Department of Biomedical Sciences for Health, University of Milan, 20133 Milan, Italy

**Keywords:** influenza, vaccination, severe acute respiratory infection, vaccine effectiveness, Italy

## Abstract

Following an extremely low incidence of influenza during the first waves of the ongoing COVID-19 pandemic, the 2021/22 Northern Hemisphere winter season saw a resurgence of influenza virus circulation. The aim of this study was to describe epidemiology of severe acute respiratory infections (SARIs) among Italian adults and estimate the 2021/22 season influenza vaccine effectiveness. For this purpose, a test-negative case-control study was conducted in a geographically representative sample of Italian hospitals. Of 753 SARI patients analyzed, 2.5% (*N* = 19) tested positive for influenza, most of which belonged to the A(H3N2) subtype. Phylogenetic analysis showed that these belonged to the subclade 3C.2a1b.2a.2, which was antigenically different from the 2021/22 A(H3N2) vaccine component. Most (89.5%) cases were registered among non-vaccinated individuals, suggesting a protective effect of influenza vaccination. Due to a limited number of cases, vaccine effectiveness estimated through the Firth’s penalized logistic regression was highly imprecise, being 83.4% (95% CI: 25.8–97.4%) and 83.1% (95% CI: 22.2–97.3%) against any influenza type A and A(H3N2), respectively. Exclusion of SARS-CoV-2-positive controls from the model did not significantly change the base-case estimates. Within the study limitations, influenza vaccination appeared to be effective against laboratory-confirmed SARI.

## 1. Introduction

Seasonal influenza carries a significant socioeconomic burden, and each year causes up to 650,000 respiratory deaths worldwide [1]. Despite vaccine effectiveness (VE) still being suboptimal [2], seasonal influenza vaccination (SIV) remains the most powerful public health means able to reduce the burden of disease and is currently recommended for several population groups, including pregnant women, older adults, young children and subjects with underlying morbidities [3].

Non-specific measures adopted during the first waves of the ongoing COVID-19 pandemic had a dramatic impact on influenza virus circulation. For instance, the 2020/21 winter season in Europe was associated with little influenza activity; indeed, a 99.8% reduction in positive detections was observed [4]. In the following 2021/22 Northern Hemisphere (NH) season influenza returned (although by a comparably small amount compared to pre-pandemic seasons) and was mostly represented by the A(H3N2) virus subtype [5]. However, the last 2022 Southern Hemisphere season (April–October 2022) was particularly severe, with influenza activity exceeding the 5-year average [6].

Monitoring of SIV VE has recently become well-established in North America, Europe and Australasia to complement existing epidemiological and virological surveillance systems [7]. This monitoring is usually performed through conducting test-negative case–control studies [7], which have an advantage of reducing differences in healthcare-seeking behavior between cases and controls [8]. Evaluation of the SIV VE may be even more important during the current COVID-19 pandemic stage. Indeed, while the initial non-specific public health measures (e.g., mask wearing and lockdowns) were able to drastically reduce influenza attack rates, these interventions have also determined an up to 60% increase in the population fraction susceptible to influenza virus [9]. These indirect effects may be substantiated by a recently report sharp increase in influenza virus circulation [6].

First available studies for the 2021/22 NH season have been conducted in an outpatient setting and suggested [10,11] a relatively low VE against A(H3N2). This was probably a consequence of predominance of the subclade 3C.2a1b.2a.2, which is antigenically different from the A(H3N2) SIV component (subclade 3C.2a1b.2a.1) [5]. In particular, the interim analysis from the United States (US) has shown that, in outpatient settings, the unadjusted and adjusted VE was 32% (95% CI: 10–50%) and 14% (95% CI: −17–37%), respectively [10]. In the Spanish region of Navarre, the observed VE was 37% (95% CI: 16–52%) against any influenza and 39% (95% CI: 16–55%) against A(H3N2) [11].

In line with other countries, the 2021/22 season in Italy was characterized by substantial influenza activity: a total of 14.5% (*N* = 13,063) of samples tested positive between November 2021 and April 2022 [12]. Considering that outpatients differ from inpatients in terms of several baseline characteristics (e.g., age and risk factors), which may affect SIV VE [13], the main goal of this hospital-based study was to analyze the epidemiology of severe acute respiratory infections (SARIs) among Italian adults and establish the 2021/22 SIV VE. Indeed, according to the World Health Organization (WHO) guidelines [8], among other influenza-related outcomes, laboratory-confirmed SARI is particularly suitable for test-negative case–control studies, as it is highly specific and may be easily integrated into the available influenza surveillance systems.

## 2. Materials and Methods

### 2.1. Overall Study Design

This study was conducted between October 2021 and May 2022 and was part of the DRIVE (Development of Robust and Innovative Vaccine Effectiveness) project, which is a multi-country network aiming to assess the brand-specific SIV VE in Europe [14]. Objectives and methods of the DRIVE project are comprehensively described at https://www.drive-eu.org/ (accessed on 14 November 2022). In the 2021/22 season, the Italian network IT-BIVE-HOSP performed a test-negative case–control study and was composed of five hospitals located in northern (regions of Liguria and Lombardy), central (regions of Tuscany and Lazio) and southern (region of Apulia) Italy.

For the 2021/22 season, SIV was recommended and fully reimbursed for older adults aged ≥60 years, pregnant women, individuals of any age with underlying medical conditions that increase their risk of influenza complications, workers operating in sectors of primary public interest (e.g., healthcare professionals, police) and some other categories [15].

### 2.2. Study Population and Procedures

All adult subjects aged ≥18 years and presenting at emergency department with SARI were potentially eligible. The per-protocol SARI definition was as follows: ≥1 systemic symptom (fever or feverishness, malaise, headache, or myalgia) or deterioration of general conditions, and ≥1 respiratory symptom (cough, sore throat, or shortness of breath) at admission or within 48 h following admission. Subjects were excluded from the analysis for the following reasons: (i) unknown or uncertain 2021/22 SIV status or SIV administered ≤14 days before SARI onset; (ii) any contraindication for SIV receipt; (iii) previous hospitalization within 48 h prior to onset of the current SARI episode or SARI onset ≥48 h following hospitalization; (iv) documented positivity to influenza before the onset of symptoms leading to the current hospital encounter; (v) institutionalized subjects.

Following the informed consent signature, samples were taken with a naso-/oropharyngeal swab that was tested for influenza types, subtypes and lineages and SARS-CoV-2 by means of reverse-transcription real-time polymerase chain reaction (RT-PCR). Cases were defined as SARI patients who tested positive for influenza, while controls were those who tested negative. At enrollment, a trained physician collected (through both interview of patient/legal representative and if possible patient’s general practitioner) demographic (sex, age, region of residence) and relevant clinical (presence of co-morbidities, current and previous SIV, and COVID-19 vaccination status) data.

### 2.3. Virus Characterization and Phylogenetic Analysis

A subset of virus detections was characterized molecularly using the Sanger method [16]. Briefly, following the RNA extraction from clinical specimens, the hemagglutinin (HA) gene of A(H3N2) viruses was amplified using the SuperScript IV One-Step RT-PCR kit (Invitrogen; Carlsbad, CA, USA) and specific primers. A nested PCR of the amplified cDNA as template was then performed. Amplification was performed using the Platinum II Taq Hot-Start DNA Polymerase (Invitrogen; Carlsbad, CA, USA). PCR products were then purified with the ExoSAP-IT PCR Product Cleanup Reagent (Applied Biosystems; Waltham, MA, USA). Finally, sequencing reactions were performed on the SeqStudio Genetic Analyzer System (Applied Biosystems; Waltham, MA, USA) for Sanger sequencing. The obtained sequences were assembled with SeqScape Software v4.0 (Applied Biosystems; Waltham, MA, USA). The nucleotide sequences generated in this study were submitted to the GISAID database (EPI_ISL_14699640, EPI_ISL_14699961, EPI_ISL_14699962, EPI_ISL_14810878, EPI_ISL_14810879, EPI_ISL_14810881).

The sequence alignment to relevant reference strains and phylogenetic analysis were performed using the “One click” workflow in the NGPhylogeny.fr software [17], which is freely accessible at https://ngphylogeny.fr (accessed on 14 November 2022).

### 2.4. Statistical Analysis

Categorical data were expressed as proportions with binomial 95% confidence intervals (CIs), while the skewed variable of age was expressed as medians with interquartile ranges (IQRs). The Fisher’s exact and Mann–Whitney–Wilcoxon’s tests were used to compare categorical and continuous variables, respectively. VE was expressed as [1 − adjusted odds ratio (aOR)] × 100% [18], and the aOR was estimated through Firth’s penalized logistic regression, which is particularly useful for rare events [19,20]. Two types of model specification were considered, namely the fully adjusted (age, sex, area of residence, month of symptom onset, time delay between the onset of symptoms and swabbing, presence of comorbidities, previous season SIV and COVID-19 vaccination pattern) and a more parsimonious specification selected by minimizing the corrected Akaike’s information criterion (AICc) [21]. We planned *a priori* to estimate VE by virus (sub)type, age-class and type of SIV.

Two kinds of sensitivity analyses were conducted. First, considering that SIV and COVID-19 vaccination behaviors may be correlated, we performed an additional de-confounding by excluding SARS-CoV-2-positive controls [22]. Second, *E*-values for both VE point estimates and 95% CIs were calculated. These latter were interpreted as the minimum strength of association that an unmeasured confounder would need to have with both the SIV receipt and positivity to influenza to fully explain away the observed association, which is conditional on the measured covariates [23].

All analyses were performed in R (v. 4.1.0) environment (R Core Team; Vienna, Austria).

## 3. Results

A total of 755 adult individuals were enrolled. Two (0.3%) subjects had an unknown 2021/22 SIV status and were excluded. In summary, a total of 753 subjects were analyzed, and their main demographic and clinical characteristics are reported in Table 1. Briefly, their median age was 73 years, and both sexes were approximately equally distributed. On median, the swab was taken 2 days after the onset of symptoms. Most (55.5%) subjects were enrolled in the southern region of Apulia, between February and March 2022 (51.5%), had no underlying health conditions (73.7%), received both the primary cycle and a booster COVID-19 vaccine dose (56.4%) and did not receive the 2020/21 SIV (59.4%). In the overall cohort, the 2021/22 SIV coverage was 42.8%. Most (62.7%; 202/322) subjects received the adjuvanted quadrivalent SIV.

During the study period, a total of 19 influenza virus detections occurred with an overall prevalence of 2.5% (95% CI: 1.5–3.9%). Eighteen (94.7%) detections were caused by A(H3N2), while the remaining case (5.3%) was a non-subtyped A virus. The HA gene of 6 out of 18 (33.3%) A(H3N2) viruses was sequenced and all belonged to the 3C.2a1b.2a.2 subclade (Figure 1. SARS-CoV-2 was detected in 283 (37.6%; 95% CI: 34.1–41.2%) samples. Moreover, there were three (0.4%; 95% CI: 0.1–1.2%) influenza and SARS-CoV-2 co-detections.

As shown in Table 2, compared with controls, cases were significantly younger and more frequently enrolled during March 2022 and in the Italian north and center. The 2021/22 SIV coverage rate was significantly (*p* = 0.004) higher in controls than in cases (43.6% vs. 10.5%), with an absolute risk reduction of 3.3% (95% CI: 1.3–5.4%).

As shown in Figure 2, when adjusted for potential confounders, the 2020/21 SIV was effective against influenza type A and A(H3N2) with point estimates of 83–84%. The corresponding *E*-values were sufficiently large, suggesting that the unmeasured confounding was unlikely to explain away the observed protective effect of SIV (Figure 2). Owing to the paucity of cases, no subgroup analysis by age-class and SIV type could be conducted.

In the sensitivity analysis, when 280 SARS-CoV-2-positive controls were excluded, no substantial changes occurred (Appendix A).

## 4. Discussion

This study showed the “return” of influenza to the Italian epidemiological scene during the 2021/22 winter season. Although most SARI cases were caused by SARS-CoV-2, a still considerable number of SARI were caused by the influenza virus. Similarly to other countries, this season in Italy was clearly dominated by the 3C.2a1b.2a.2 subclade of the A(H3N2) virus subtype, which is antigenically different from the 2021/22 NH SIV A(H3N2) component A/Cambodia/e0826360/2020 (subclade 3C.2a1b.2a.1) [5,24]. Despite this apparent mismatch, in our study, most (89.5%; 17/19) SARI cases were not vaccinated, suggesting a significant protection provided by SIV. In this regard, a systematic review and meta-analysis of adult randomized controlled trials by Tricco et al. [25] has reported a similar vaccine efficacy against both matched (61%; 95% CI: 46–73%) and mismatched (64%; 95% CI: 23–82%) influenza A virus.

As compared with our findings and despite similarities in virus circulation, the available US [10] and Spanish [11] estimates for the 2021/22 VE are substantially lower (14–39%). This apparent inconsistency may be explained by a number of concurrent reasons. First, both US [10] and Spanish [11] studies were conducted in the outpatient setting, and therefore cases were not severe. In this regard, Tendorfe et al. [13] have documented that hospital-based and community-based SIV VE studies are composed of two distinct populations, and in some instances, the VE computed in an inpatient setting may be higher than among outpatients. Analogously, the pooled analysis performed within the DRIVE project showed that the VE among working-aged adults was 0% (95% CI: −96–49%) and 85% (95% CI: 12–97%) in primary-care-based (data from four networks located in Austria, Iceland, Italy and United Kingdom) and hospital-based (data from nine networks located in France, Iceland, Italy, Romania and Spain) test-negative studies, respectively [26]. In summary, it is highly plausible that SIV VE may be higher against more severe clinical outcomes, as it may reduce the severity of disease in cases where it did not prevent infection per se [27]. Second, in our study, two-thirds of vaccinated subjects received the adjuvanted SIV. Bolton et al. [24] have recently shown that antibodies elicited by the 2021/22 NH non-adjuvanted standard-dose egg-based SIV poorly neutralized strains belonging to the 3C.2a1b.2a.2 A(H3N2) subclade. By contrast, the MF59 adjuvant may broaden the immune response beyond the strains included in SIV [28]. Indeed, the adjuvanted formulation has been extensively shown to be superior to non-adjuvanted SIVs in inducing heterologous and cross-clade immune responses, particularly within the A(H3N2) subtype [29,30,31,32]. Analogously, our recent study [33] demonstrated that during the 2018/19 and 2019/20 seasons, which were characterized by a significant proportion of drifted A(H3N2) strains, the adjuvanted SIV was 59.2% (95% CI: 14.6–80.5%), more effective than non-adjuvanted comparators. Finally, it should be kept in mind that our effect sizes were imprecise, and the previously reported VE point estimates [10,11] were usually within the 95% CIs reported in the present report.

Interestingly, SARI patients positive for influenza were significantly younger than those positive for SARS-CoV-2. Similar observations have been reported from other studies carried out in the United Kingdom [34] and Brazil [35]. In Italy, older age is among the most important predictors of both SIV [36] and COVID-19 [37] vaccination uptake. On considering this positive relationship between SIV and COVID-19 vaccination, in test-negative case–control studies, a bias driven by the conditional probability of vaccination may be introduced [22]. It seems that this bias may be particularly important for older adults. Although in our sensitivity analysis (in which SARS-CoV-2-positive SARI patients were excluded), no major changes in the SIV VE took place, the limited number of cases did not allow us to assess age-specific estimates. Further studies on this purpose are warranted.

As we mentioned earlier, the main drawback of the present study is a limited number of cases imposed by a lower than average circulation of influenza. This limitation determined both the imprecise VE estimates with large 95% CIs and the impossibility of performing further subgroup analyses. We tried to limit the impact of this limitation at the phase of data analysis by applying Firth’s penalized estimation methods able to provide bias-reduced regression coefficients in datasets with rare events [38]. Moreover, although the calculated *E*-values were sufficiently large and suggested that a relatively strong unmeasured confounding was needed to negate our estimates, we cannot definitively rule out the issue of residual confounding.

In conclusion, the 2021/22 winter season in Italy saw a resurgence of influenza virus circulation, and a substantial proportion of SARI cases were caused by influenza A(H3N2). These latter were part of the 3C.2a1b.2a.2 subclade, which was antigenically different from the A(H3N2) component of the 2021/22 NH SIV. Despite vaccine mismatch, most laboratory-confirmed SARI cases were recorded among non-vaccinated inpatients, suggesting a protective effect of the 2021/22 NH SIV. Effective public health interventions to increase SIV uptake (with the updated 2022/23 NH SIV formulation containing a 3C.2a1b.2a.2-like virus) in all principal target groups [3] should be pursued.

## Figures and Tables

**Figure 1 vaccines-11-00083-f001:**
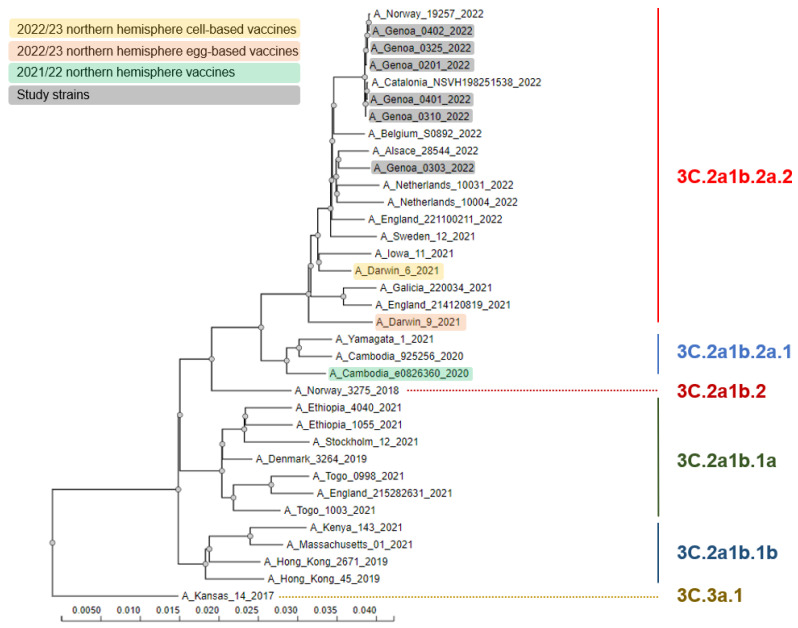
Phylogenetic comparison of influenza A(H3N2) hemagglutinin genes identified in the study.

**Figure 2 vaccines-11-00083-f002:**
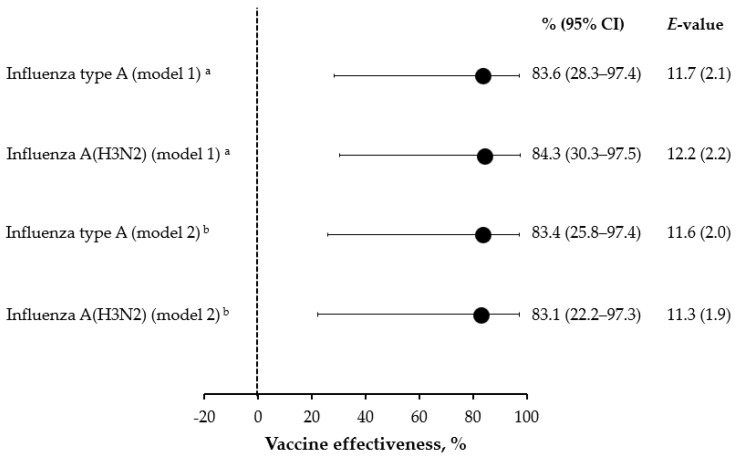
Effectiveness of the 2021/22 seasonal influenza vaccination against severe acute respiratory infection. ^a^ Covariates (age, area, month of symptom onset, COVID-19 vaccination pattern and previous season influenza vaccination) selected on the basis of minimization of the corrected Akaike’s information criterion; ^b^ Fully adjusted (age, sex, area, month of symptom onset, swab delay, presence of chronic conditions, COVID-19 vaccination pattern, and previous season influenza vaccination) model.

**Table 1 vaccines-11-00083-t001:** Characteristics of the study participants (*N* = 753).

Characteristic	Level	% (*N*)
Age, years	Median (IQR)	73 (58–83)
Sex	Female	46.3 (349)
	Male	53.7 (404)
Area	North	13.0 (98)
	Center	31.5 (237)
	South	55.5 (418)
Month of symptom onset	Oct–Nov 2021	6.0 (45)
	Dec 2021	7.2 (54)
	Jan 2022	20.6 (155)
	Feb 2022	24.3 (183)
	Mar 2022	27.2 (205)
	Apr–May 2022	14.7 (111)
Swab delay, days	Median (IQR)	2 (0–4)
≥1 chronic condition	No	73.7 (555)
	Yes	26.3 (198)
COVID-19 vaccination	No	14.3 (108)
	1 dose	3.7 (28)
	2 doses	25.5 (192)
	3 doses	56.4 (425)
2020/21 influenza vaccination	No	59.4 (447)
	Yes	32.1 (242)
	Unknown	8.5 (64)
2021/22 influenza vaccination	No	57.2 (431)
	Yes	42.8 (322)

**Table 2 vaccines-11-00083-t002:** Comparison of cases and controls.

Characteristic	Level	Cases (*N* = 19)	Controls (*N* = 734)	*p*
Age, years	Median (IQR)	43 (28–69)	73 (58–83)	<0.001 ^a^
Sex	Female, % (*N*)	47.4 (9)	46.3 (340)	>0.99 ^b^
	Male, % (*N*)	52.6 (10)	53.7 (394)	
Area	North, % (*N*)	42.1 (8)	12.3 (90)	<0.001 ^b^
	Center, % (*N*)	47.4 (9)	31.1 (228)	
	South, % (*N*)	10.5 (2)	56.7 (416)	
Month of symptom onset	Oct–Nov 2021, % (*N*)	0 (0)	6.1 (45)	0.003 ^b^
	Dec 2021, % (*N*)	10.5 (2)	7.1 (52)	
	Jan 2022, % (*N*)	5.3 (1)	21.0 (154)	
	Feb 2022, % (*N*)	5.3 (1)	24.8 (182)	
	Mar 2022, % (*N*)	68.4 (13)	26.2 (192)	
	Apr–May 2022, % (*N*)	10.5 (2)	14.9 (109)	
Swab delay, days	Median (IQR)	3 (0–4)	2 (0–4)	0.84 ^a^
≥1 chronic condition	No, % (*N*)	73.7 (14)	73.7 (541)	>0.99 ^b^
	Yes, % (*N*)	26.3 (5)	26.3 (193)	
COVID-19 vaccination	No, % (*N*)	0 (0)	14.7 (108)	0.092 ^b^
	1 dose, % (*N*)	10.5 (2)	3.5 (26)	
	2 doses, % (*N*)	31.6 (6)	25.3 (186)	
	3 doses, % (*N*)	57.9 (11)	56.4 (414)	
2020/21 influenza vaccination	No, % (*N*)	63.2 (12)	59.3 (435)	0.063 ^b^
	Yes, % (*N*)	15.8 (3)	32.6 (239)	
	Unknown, % (*N*)	21.1 (4)	8.2 (60)	
2021/22 influenza vaccination	No, % (*N*)	89.5 (17)	56.4 (414)	0.004 ^b^
	Yes, % (*N*)	10.5 (2)	43.6 (320)	

IQR, Interquartile range; ^a^ Mann–Whitney–Wilcoxon test; ^b^ Fisher’s exact test.

## Data Availability

Further details and results from the DRIVE studies are available at the DRIVE website via the following link: https://www.drive-eu.org/ (accessed on 14 November 2022).

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
