# Peer review of "Surveillance of Severe Acute Respiratory Infection and Influenza Vaccine Effectiveness among Hospitalized Italian Adults, 2021/22 Season"

_vaccines, 2022, doi:10.3390/vaccines11010083_

Round 1
Reviewer 1 Report
I think this is a decent study, I enjoyed it.
---------updated
The study focused on SARS-COV-2 positivity and influenza positivity among a group of SARI patients across late 2021 and early 2022. These data show vivid competition of the two virus, and the study estimated the protective effects of influenza vaccine.
This is timely study, and on an important questions, e.g. the competition and co-circulation of two respiratory viruses, and the efficacy of the influenza vaccine. The only limitation is the small sample size, but the authors had discuss this limitation.
I have major comments. Maybe more discussion on the infection of previous COVID-19 infection against reinfection and influenza infection. The group includes most elderly.
Author Response
Comment: I think this is a decent study, I enjoyed it. The study focused on SARS-COV-2 positivity and influenza positivity among a group of SARI patients across late 2021 and early 2022. These data show vivid competition of the two virus, and the study estimated the protective effects of influenza vaccine. This is timely study, and on an important questions, e.g. the competition and co-circulation of two respiratory viruses, and the efficacy of the influenza vaccine. The only limitation is the small sample size, but the authors had discuss this limitation.
Reply: Thank you for your interest in our paper. All your comments have been addressed.
Comment: I have major comments. Maybe more discussion on the infection of previous COVID-19 infection against reinfection and influenza infection. The group includes most elderly.
Reply: As suggested, we have now discussed more in detail (with a particular emphasis on age) the comparison between SARI caused by influenza and SARS-CoV-2.
Reviewer 2 Report
The study, aimed to analyze epidemiology of severe acute respiratory infection and assess effectiveness of influenza vaccine in the 2021/22 season among in Italian adults, is well designed and described.
The study was conducted in the hospital settings, which is probably (as was underlined in the Discussion) a cause of higher vaccine effectiveness than reported by others. So please consider a change in the title e.g. "...among hospitalized Italian adults".
The influenza vaccination coverage among the study participants in the 2021/22 season was 42,8%. It would be worth to provide information if influenza vaccine is recommended for all adults and its cost is partially or fully reimbursed in Italy.
The most important limitation of the study is a small number of influenza cases. The implications were discussed in detail. Although further analyses were therefore not possible, the reported observations are worth attention.
Author Response
Comment: The study, aimed to analyze epidemiology of severe acute respiratory infection and assess effectiveness of influenza vaccine in the 2021/22 season among in Italian adults, is well designed and described.
Reply: Thank you for your interest in our paper. All your comments have been addressed.
Comment: The study was conducted in the hospital settings, which is probably (as was underlined in the Discussion) a cause of higher vaccine effectiveness than reported by others. So please consider a change in the title e.g. "...among hospitalized Italian adults".
Reply: The title has been changed accordingly.
Comment: The influenza vaccination coverage among the study participants in the 2021/22 season was 42,8%. It would be worth to provide information if influenza vaccine is recommended for all adults and its cost is partially or fully reimbursed in Italy.
Reply: As required, we have now added a description of the Italian influenza vaccination policies, including recommendations and reimbursement policies.
Comment: The most important limitation of the study is a small number of influenza cases. The implications were discussed in detail. Although further analyses were therefore not possible, the reported observations are worth attention.
Reply: We agree that a relatively small number of influenza cases is the main study limitation. We tried to address this shortcoming by applying robust and penalized regression models. In any case, we have clearly indicated this limitation.
Reviewer 3 Report
"The aim of this study was to describe epidemiology of severe acute respiratory infection 28 (SARI) among Italian adults and estimate the 2021/22 season influenza vaccine effectiveness." The main problem in the study design is the recruitment of cases and controls via routine diagnostics. As a result, cases and controls are very unbalanced in essential factors. However, since the balance of cases and controls is essential for such a study design, a serious recruitment BIAS can not be excluded, so all conclusions must be questioned. In conclusion the study can not be recommended for publication.
Author Response
Protection induced by influenza vaccines may be measured in both randomized controlled trials (which measure the vaccine efficacy) and observation studies (which measure the vaccine effectiveness). Continuous monitoring of the influenza vaccine effectiveness is recommended by several regulatory agencies, including the WHO. In observational studies, baseline imbalances are frequent and therefore statistical adjustment to establish the casual relationship is necessary. As per WHO guidelines (https://apps.who.int/iris/bitstream/handle/10665/255203/9789241512121-eng.pdf), laboratory-confirmed SARI is a very specific influenza-related outcome and, as such, recommended for test-negative case-control studies. In the same way, the test-negative case-control design reduces confounding due to differences in health-care seeking behavior between cases and non-cases, which is a major challenge to influenza VE studies. In the present study, the eventual selection and confounding by indication biases were accounted for in a robust multivariable logistic model. Moreover, an extensive sensitivity analysis was conducted to confirm robustness of the base-case. Principal study limitations have been discussed.
Reviewer 4 Report
The goal of this study was to explain the epidemiology of severe acute respiratory infection (SARI) among Italian adults and estimate the effectiveness of the influenza vaccine for the 2021/22 season. A test-negative case-control study was conducted in a geographically representative sample of Italian hospitals for this purpose. The manuscript is well-structured and well-discussed. However, some points should be checked and corrected before it's accepted in this journal.
Therefore, according to my comments, I recommended the publication of the paper after minor revision.
[1] The study's background should be clearly stated. Describe the introduction and review of the work (Please add more information).
[2] Please speculate on the results. The discussion must improve.
[3] The MS English needs to be improved. The article's English must be carefully checked for grammatical errors.
Author Response
Comment: The goal of this study was to explain the epidemiology of severe acute respiratory infection (SARI) among Italian adults and estimate the effectiveness of the influenza vaccine for the 2021/22 season. A test-negative case-control study was conducted in a geographically representative sample of Italian hospitals for this purpose. The manuscript is well-structured and well-discussed. However, some points should be checked and corrected before it's accepted in this journal. Therefore, according to my comments, I recommended the publication of the paper after minor revision.
Reply: Thank you for your interest in our paper. All your comments have been addressed.
Comment: [1] The study's background should be clearly stated. Describe the introduction and review of the work (Please add more information).
Reply: As suggested, we have now enriched the Introduction section by substantiating the study background.
Comment: [2] Please speculate on the results. The discussion must improve.
Reply: We have now revised the whole discussion section. Several relevant studies have been added, discussed and compared with our findings.
Comment: [3] The MS English needs to be improved. The article's English must be carefully checked for
Reply: The whole manuscript has been now revised by a native speaker.
Round 2
Reviewer 3 Report
The authors use a test-negative case-control study with a ratio of cases to controls of 1:38; in total, only 19 cases were used for the effect estimates. Cases and controls differ greatly from each other, especially in age (medians 43 (cases), 73 (controls)). The seasons of symptom onset are also very differently distributed in both groups. In this respect, the authors refer to the performance of appropriately adjusted analyses. However, even adjusted analyses can only partially eliminate weaknesses in the study design. The extremely small number of cases, combined with large differences in the structure of cases and controls, reveal a lack of care in the study design, so that the results are potentially highly biased. These deficiencies in the study design contradict the general conclusions of the authors. Acceptance of this paper is not recommended due to the methodological deficiencies in study design.
Author Response
We do not agree with this comment. First of all, the study design is appropriate, the DRIVE is a pan-European project, whose design has been established by several experts. The limited number of cases is outside researchers’ control. Indeed, it is well-known that non-specific SARS-CoV-2 interventions reduced circulation of influenza. Moreover, this study shortcoming was extensively described among the study limitations.
Second, vaccine effectiveness in case-control studies may be established even for very small number of cases. See, for example, this paper published in JAMA (https://jamanetwork.com/journals/jama/fullarticle/187226), where a total of 17 cases and 84 controls were used to estimate the vaccine effectiveness. The small number of cases MAY determine wide confidence intervals; this fact of THE imprecise estimate has been highlighted in both the abstract and discussion. Finally, the post-hoc calculation (see the file attached) of the study power suggest that our study had an 84.7% power.
